# A Review of Neovascular Glaucoma: Etiology, Pathogenesis, Diagnosis, and Treatment

**DOI:** 10.3390/medicina58121870

**Published:** 2022-12-18

**Authors:** Danielė Urbonavičiūtė, Dovilė Buteikienė, Ingrida Janulevičienė

**Affiliations:** 1Medical Academy, Lithuanian University of Health Sciences, Eiveniu 2, LT-50161 Kaunas, Lithuania; 2Department of Ophthalmology, Medical Academy, Hospital of Lithuanian University of Health Sciences Kaunas Clinics, LT-50161 Kaunas, Lithuania

**Keywords:** neovascular glaucoma, retinal ischemia, neovascularization, glaucoma surgery, intravitreal anti-VEGF therapy

## Abstract

Neovascular glaucoma (NVG) is a rare, aggressive, blinding secondary glaucoma, which is characterized by neovascularization of the anterior segment of the eye and leading to elevation of the intraocular pressure (IOP). The main etiological factor is retinal ischemia leading to an impaired homeostatic balance between the angiogenic and antiangiogenic factors. High concentrations of vasogenic substances such as vascular endothelial growth factor (VEGF) induce neovascularization of the iris (NVI) and neovascularization of the angle (NVA) that limits the outflow of aqueous humor from the anterior chamber and increases the IOP. NVG clinical, if untreated, progresses from secondary open-angle glaucoma to angle-closure glaucoma, leading to irreversible blindness. It is an urgent ophthalmic condition; early diagnosis and treatment are necessary to preserve vision and prevent eye loss. The management of NVG requires the cooperation of retinal and glaucoma specialists. The treatment of NVG includes both control of the underlying disease and management of IOP. The main goal is the prevention of angle-closure glaucoma by combining panretinal photocoagulation (PRP) and antiangiogenic therapy. The aim of this review is to summarize the current available knowledge about the etiology, pathogenesis, and symptoms of NVG and determine the most effective treatment methods.

## 1. Introduction

Glaucoma is a chronic eye disease damaging the optic nerve and leading to permanent blindness globally [1]. The number of people with glaucoma worldwide has increased from 64.3 million in 2013, and by 2040, it is expected to reach over 111 million [2]. NVG is a vision-threatening secondary glaucoma characterized by neovascularization of the iris associated with IOP elevation [3]. This glaucoma was first documented in 1871 and has been mentioned as congestive glaucoma, rubeotic glaucoma, or diabetic hemorrhagic glaucoma. Historically, the term neovascular glaucoma appeared when, in 1906, Coats first described the histopathological findings of new blood vessels in a patient with central retinal vein occlusion (CRVO), and in 1928, Salus discovered similar blood vessels in the iris of a diabetic patient. Finally, in 1963, Weiss et al. used the term “neovascular glaucoma”, for the first time associating it with neovascularization of the iris (rubeosis), fibrovascular membrane in the corner of the anterior chamber, and elevated IOP [4,5]. 

NVG is divided into three clinical stages: preglaucoma, open angle glaucoma, and angle-closure glaucoma, which can be determined by anterior biomicroscopy and gonioscopy [3]. The pre-rubeotic stage can be detected in the early stage of NVG, when there are no symptoms and clinical findings are related to underlying diseases causing ocular ischemia, such as proliferative diabetic retinopathy (PDR), CRVO, and ocular ischemic syndrome (OIS) [6]. Despite the fact that there are many different systemic and ocular diseases inducing the development of NGV, they all share the same etiological factor: retinal ischemia and chronic hypoxia (97%); the other 3% includes inflammatory diseases and intraocular tumors [7,8]. Retinal ischemia stimulates the release of VEGF, which induces angiogenesis. Under normal conditions, it plays an important angiogenic role: promotes the growth of new blood vessels. When an excessive amount of VEGF is released, it promotes the formation of pathological blood vessels in the iris and in the iridocorneal angle [9]. 

NVG treatment includes not only reducing high IOP with topical anti-glaucoma drugs and surgery but also inhibiting the main mechanism of NVG development using PRP and intravitreal anti-VEGF injections. However, in most cases, pharmacological treatment does not provide an adequate response of elevated IOP; therefore, surgical interventions such as drainage valve implantation, trabeculectomy with mitomycin C (MMC) or 5-Fluorouracil (5-FU), and cyclodestructive procedures are usually required [10]. Although trabeculectomy is considered to be the most common surgical option in patients with NVG due to its success rate, Kentaro Iwasaki el al. demonstrated that Baerveldt glaucoma implant surgery achieved a higher success rate than trabeculectomy when IOP was monitored between 21 and 17 mmHg [11,12]. 

To stop the progression of NVG, an early diagnosis is necessary, because this disease has great importance in the damage to the optic nerve, which results in permanent visual impairment. In patients with NVG, an advanced stage of the disease is often observed, so the treatment of this pathology remains complicated. 

We provided a literature search. A systematic review of the literature was performed, and scientific articles were selected according to the inclusion and exclusion criteria. Computer bibliographic medical databases PubMed (MEDLINE^®^), Science direct (EMBASE^®^), Web of Science^®^, and Cochrane Library^®^ were used as the data sources. Keywords in variuos combinations were used for the literature search: “neovascular glaucoma”, “glaucoma drainage devices”, “trabeculectomy”, “retinal ischemia”, “anti-VEGF intravitreal therapy”, “topical antiglaucoma medications”, “cyclodestructive procedures”, and “pan-retinal photocoagulation”. Inclusion criteria: patients with NVG; publications with NVG surgical methods and conservative treatment; publications describing NVG etiology, pathogenesis, and diagnostics; articles dated between 1963 and 2022; randomized controlled trials (RCTs), literature reviews, and meta-analyses. Exclusion criteria: publications not written in English; publications older than 1963. 

## 2. Prevalence and Etiology

NVG is a rare pathology, and the prevalence in the population reaches only 0.01–0.12%. The incidence of NVG is 3.9% of all glaucoma cases (9–14.7% of all secondary glaucomas), and it increases with the prevalence of diabetics with PDR [4,13]. In diabetic patients with poor glycemic control and advanced untreated posterior segment ischaemia, the progression of NVG may occur after 12 months since the development of the NVI [14]. According to the recent Global Burden of Disease Study analysis, in 2020, DR was one of the main causes of NVG and the 5^th^ disorder initiating blindness in those aged over 50 years [15]. In patients with a retinal ischemic event, NVG usually develops within 1.5 to 6 months [16]. A clinical study in a tertiary Brazilian hospital found that, although the incidence of NVG was similar between the genders, it remained slightly higher in the male population. Of the 38 patients studied, 46.16% were between 60 and 79 years, and 30.68% were over 80 years, suggesting that older people are more likely to develop NVG [17]. 

Systemic diseases such as diabetes, hypertension, and hyperlipidemia can result in NVG, but despite all these pathologies, there are three main ophthalmological causes, including PDR, CRVO, and ocular ischemic syndrome [18]. Potential causes that may influence NVG can be divided into several groups (Table 1). 

## 3. Pathogenesis

NVG is a tremendous ocular condition caused by new blood vessels in the iris and in the anterior chamber corner, forming in response to retinal ischemia and leading to uncontrolled IOP rise [6]. Ischemic retina produces vasogenic substances that stimulate the growth of new vessels. The pathological process of angiogenesis is due to an impaired homeostatic balance between angiogenic factors, such as vascular endothelial growth factor (VEGF); hepatocyte growth factor; insulin-like growth factor; tumor necrosis factor; inflammatory cytokines (especially IL-6); and antiangiogenic factors: transforming growth factor (TGF-β), thrombospondin, and somatostatin [23]. 

VEGF is one of the main vascular permeability factor that plays a crucial role in neovascularization. VEGF is released by various retinal cells, including pericytes, retinal pigment epithelium, nonpigmented ciliary epithelium, Muller, and ganglion cells [24]. In NVG, excessive amounts of VEGF enter the anterior chamber of the eye from the posterior pole and induce the development of neovascularization (NV), initially from the capillaries of the iris minor and major arterial rings and then further spreading to the anterior chamber angle. When the extremely high levels of VEGF are released, it causes the breakdown of the blood–retinal barrier by increasing the leukocyte adhesion to endothelial cells [25,26]. TGF-β and fibroblast growth factors are responsible for fibroblast proliferation and fibrovascular membrane formation over the iris and the angle of the anterior chamber [27,28]. This membrane with proliferated myofibroblasts can cause an obstruction, which decreases the drainage of the aqueous outflow through the spongy tissue between the cornea and the sclera called trabecular meshwork (TM), leading to secondary open-angle stage of NVG. Eventually, the contraction of the fibrovascular membrane occurs, causing the angle-closure stage. This causes an increase in the intraocular pressure (IOP), which can progress to glaucomatous optic neuropathy (GON) and result in irreversible blindness [18]. Therefore, the early detection and treatment of NVG is important (Figure 1).

## 4. Clinical Profile

Chronically red and painful eyes are the first visible clinical symptoms of NVG. Although, in young patients, it could be asymptomatic due to an endothelial function reserve [29]. Some patients may experience light sensitivity or blurred vision [30]. Secondary angle-closure NVG develops more serious symptoms, including severe eye pain, headaches associated with nausea, and vomiting [31]. 

The clinical features include visible NVI and NVA. In rare cases, NVA could possibly be seen without NVI. Usually, the pupillary margin is the place where NVI begins, but it may also appear on the surface or in periphery of the iris and at a YAG iridotomy. Increased IOP ranges from 40 to 60 mmHg and higher. Elevated IOP leads to corneal edema [3]. Gonioscopically, partial or complete angle-closure caused by the fibrovascular membrane is observed. It is important to mention that, in 10% of CRVO cases, newly formed blood vessels will initially occur in the anterior chamber angle and only later in the iris itself, so gonioscopy should not be forgotten [32]. Changes in the fundus of the eye are associated with clinical features of other retinal disorders, such as DR, CRVO, or others [3,4]. The clinical stages of NVG are shown in Table 2.

## 5. Ophthalmic Diagnostics

Two main methods are used to identify NVG: slit-lamp biomicroscopy of the anterior segment of the eye and gonioscopy (Figure 1a). In the early stages, the NVI may be unnoticeable, so it can be detected by iris fluorescein angiography (FAG) [33]. FAG is considered to be the gold standard in identifying NV and ischemic areas in the eye blood circulation (Figure 1b) [34,35]. This method is used to determine retinal and choroidal blood flow with fluorescein dyes. FAG evaluates the blood vessels, the presence of NV, circulatory disorders, or increased vascular permeability. The early sign of NV in the iris is the leakage of intravenously injected sodium fluorescein dye from vessels at a pupillary margin [3]. This is the pre-rubeosis phase, when the NVI and NVA are not clinically visible by slit-lamp biomicroscopy but may be observed at the pupillary edge of the iris by FAG. During vascular proliferation, the pre-rubeosis phase transitions to rubeosis, so it can already be seen by slit-lamp biomicroscopy. During the slit-lamp examination, the NVI vessels appear as thin, tortuous, and irregularly arranged on the surface of the iris, and they must be differentiated from the normal radial vessels from the ciliary trunk and the radial iris vessels from the circular ciliary band [18]. When the NVI is noticed, careful gonioscopy should be performed to identify the NVA and determine any existing fibrovascular membranes or peripheral anterior synechiae. Retinal ischemia and NV can also be assessed by optical coherent tomography angiography (OCTA (Figure 1c) [36]. 

In advanced stages, NVG can be associated with a rare eye condition called Ectropion Uveae (EU). EU is often observed when altered vessels and the contracting fibrovascular membrane pulls the pupillary margin, causing movement of the iris pigment epithelium on the anterior surface of the iris [37]. 

NVG patients should be carefully investigated because of the systemic associations:If diabetes is suspected, blood glucose levels and glycated hemoglobin (HbA1c) should be measured [3].To determine arterial hypertension, it is important to regularly measure arterial blood pressure [18,38].In order to avoid ishemic eye syndrome, it is necessary to perform carotid artery doppler (retrobulbar, intra-, and extracranial vessels); magnetic resonance imaging (MRI); and computed tomography (CT) with angiography [6,39].Carotid-cavernous fistula and tumor metastases could be excluded by MRI, CT, and positron emission tomography (PET) [3,39].For uveitis, retinal vasculitis, and blood pathology (dyscrasias), the HLA B27 complex should be tested [4].Specific tests should be performed to exclude tuberculosis (QuantiFERON-TB Gold, Mantoux) and sarcoidosis (ACE, chest X-ray) [3,18,30].

## 6. Treatment 

The management of NVG may be challenging and frustrating, often requiring the cooperation of retinal and glaucoma specialists. According to European Glaucoma Society, the primary key principles of NVG management are divided into two different directions: (1) the treatment of retinal ischemia by performing pan-retinal photocoagulation (PRP) or intravitreal anti-VEGF injections and (2) controlling high IOP by topical and systemic pharmacological medications or surgical interventions to avoid damage of the optic nerve [40,41].

### 6.1. Treating the Underlying Pathology

The most common causes of NVG are ischemic eye diseases such as PDR, CRVO, and OIS. PRP is currently the most common treatment to minimize posterior segment retinal ischemia (Figure 2) [42]. The goal of this procedure is to reduce the ischemic areas in the retina by increasing the oxygen supply and, also, to reduce the secretion of VEGF and interleukin-6. The typical photocoagulation parameters for NVG are 1200–1600 burns, approximately 500 micron spots in size. Laser procedures are performed in 1–3 sessions in a period of 5–7 days, and it should be done as fast as possible [20]. In certain cases, when the patient cannot undergo PRP due to cloudy media, cryotherapy or pars plana vitrectomy (PPV) + endolaser are used for retinal detachment [43]. Furthermore, in severe conditions, triple therapy may be required: PPV + endolaser or PRP associated with anti-VEGF injection [18]. 

The currently available anti-VEGF inhibitors, including bevacizumab (Avastin), ranibizumab (Lucentis), and aflibercept (Eylea), have proven to be effective suppressing anterior segment neovascularization and lowering IOP. 

Bevacizumab (Avastin)—a full-length humanized, recombinant monoclonal IgG antibody that inactivates all VEGF isoforms [44]. Asaad A. Ghanem et al. reported a significant regression of iris neovascularization and IOP reduction (from 28 ± 9.3 mm Hg to 21.7 ± 11.5 mm Hg) 1 week after intravitreal bevacizumab injection in eyes with NVG. This was a prospective, observational clinical case series study in which 16 patients received an intravitreal injection of bevacizumab (2.5 mg) and followed for 2 months. Partial or complete regression of NVI was observed 1 week after the injection [45]. 

Jun Young Ha et al. evaluated the efficacy of an intracameral bevacizumab injection in patients with NVG. This was a retrospective study that included 26 eyes of 26 NVG patients who received intracameral bevacizumab (25 mg/mL) injection and were followed up for 1 year. Additional treatment was also performed: all patients were treated with topical or systemic IOP-lowering medications and PRP. After 1 week, an intracameral injection of bevacizumab resulted in a significant reduction of IOP (baseline: 39.79 ± 5.33 mmHg; 1 week after: 16.5 ± 3.4 mmHg), NVI, and NVA. However, in 4 out of 26 eyes, the IOP-lowering effect was insufficient, so additional surgical treatment was performed 1 week after injection. Lastly, 12 months after the injection, the number of eyes requiring glaucoma surgery increased to 19. In conclusion, intracameral bevacizumab is an effective way to reduce IOP and to stop the regression of neovascularization; unfortunately, the therapeutic effect is transient, and some patients had to receive surgery such as trabeculectomy with mitomycin C, Ahmed valve implantation, or transscleral cyclophotocoagulation [46]. 

Ranibizumab (Lucentis)—recombinant, humanized, antigen-binding fragment of monoclonal antibodies that binds and neutralizes all biologically active forms of VEGF. It has a high affinity for VEGF-A, inhibiting its action and blocking the stimulation of the VEGFR1 and VEGFR2 receptors [47]. De-Kun Li et al. investigated the clinical efficiency of ranibizumab combined with surgical treatment. In this study, they included 11 patients (13 NVG-affected eyes) injected with a single intravitreal dose of 0.5 mg ranibizumab one week before trabeculectomy. The results showed that the intravitreal injection of ranibizumab combined with trabeculectomy effectively reduced the IOP (preoperative IOP was 48.5 ± 11.76 mmHg; postoperative IOP 1 week after 20.88 ± 4.91 mmHg; 1 month after was 19.71 ± 2.69 mmHg) [48]. Julia Lüke et al. presented a prospective, monocenter, 12-month, interventional case series study in which 20 patients were enrolled with rubeosis and NVG. All eyes were injected intravitreally at the baseline, then, if needed, once monthly. The authors‘ results suggested that an intravitreal injection of ranibizumab (0.5 mg) is beneficial as an adjuvant therapy for patients with NVG [49]. 

Aflibercept (Eylea)—composed of the core domains of human IgG1 with an approximately 100-fold higher binding affinity for VEGF-A than ranibizumab [50]. Masaru Inatani et al. performed a randomized, double-masked, sham-controlled, phase 3 trial in Japan (VEGA study) and investigated the effect of intravitreal aflibercept (IVT-AFL) monotherapy in patients with NVG. Fifty-four patients were enrolled with anterior segment neovascularization and IOP > 25 mmHg. All eyes were randomly allocated and received background therapy plus IVT-AFL (2mg) or a sham injection at the baseline. At week 1, the authors concluded that IVT-AFL resulted in a significant reduction of the IOP (mean change was −9.9 mmHg) that, in the sham group, was −5.0 mmHg. The improvement in grade of the NVI was also detected [51]. 

Brolucizumab (Beovu)—fragment of a humanized monoclonal single-chain Fv antibody. A significantly higher binding affinity of brolucizumab to VEGF-A isoforms was observed compared to bevacizumab. The smaller molecular size of brolucizumab provides potentially faster retinal penetration compared to other VEGF inhibitors [52]. In 2019, brolucizumab was approved by the Food and Drug Administration (FDA) for the treatment of neovascular (wet) age-related macular degeneration (AMD) and diabetic macular edema (DME). Currently, there are no studies confirming the effectiveness of brolucizumab in the treatment of NVG. 

These medications cause a rapid regression of NV in the anterior chamber and reduce the IOP within a few days [20,53]. It is an alternative drug therapy for NVG when a PRP is not possible because of cloudy media. Although the effect of anti-VEGF drugs for treating patients with NVG is temporary and lasts 1-1.5 months, so many studies suggest combining anti-VEGF therapy with PRP or surgical treatment [54,55,56]. The short-term IVT anti-VEGF therapy is beneficial, but currently there are insufficient data on the long-term efficacy of IVT anti-VEGF as an adjunct to conventional IOP-lowering therapy in NVG. Therefore, IVT anti-VEGF is not yet recommended as a routine treatment in NVG. In urgent cases, intravitreal injections can be a beneficial treatment for decreasing intraoperative bleeding from fragile vessels. 

### 6.2. The Management of Intraocular Pressure

#### 6.2.1. Pharmacological Treatment

The management of IOP often requires both topical and systemic treatment, but, in most cases, surgical intervention is necessary. The main aqueous suppressants, such as topical β-blockers, α-agonists, and carbonic anhydrase inhibitors, may be beneficial in reducing the IOP [4]. Systemic carbonic anhydrase inhibitors can also be effective for temporary short-term IOP reduction but must be used carefully in patients with impaired renal function [30]. Topical corticosteroid (Dexamethasone) therapy reduces inflammation, vascular permeability, and new vessel growth, and cycloplegic medications (Atropine) reduce pain.

Not recommended:(1)Prostaglandins—increases fluid leakage through trabecular and uveoscleral pathways, so their effectiveness is questionable; contraindicated when the inflammation is observed.(2)Anti-cholinergics are contraindicated because of an increase in inflammation and also cause miosis due to further forward movement of the iris lens diaphragm, increasing the anterior chamber angle-closure and reducing uveoscleral leakage [20].

#### 6.2.2. Surgical Treatment

NVG is a refractory form of secondary glaucoma in which adequate control of intraocular pressure is difficult and is often associated with an increased risk of postoperative complications such as hyphema and vision loss. Although the main treatment of patients with NVG is the reduction of retinal ischemia with PRP, it is often necessary to control increased IOP with surgical interventions in order to prevent irreversible complications such as optic nerve damage. Approximately 50% of eyes with NVG require surgical IOP control [57].

Indications for the surgical treatment of NVG:valuable visual acuity (VA) >0.05;insufficient maximum pharmacological control of IOP: >50% of NVG patientsclosed anterior chamber angle due to synechiae.

The following surgical procedures are: trabeculectomy with or without antimetabolites, the implantation of aqueous drainage devices, cyclodestructive procedures (cyclophotocoagulation and cyclocryotherapy), and PPV + endolaser and glaucoma drainage devices [58].

Trabeculectomy (Glaucoma filtration surgery) is currently the most common surgical treatment option for NVG. Trabeculectomy is indicated when medical and/or laser treatments do not provide sufficient IOP control or disease progression is observed [59]. The treatment involves creating a fistula that connects the anterior chamber of the eye with the subconjunctival space, thus creating an alternative drainage path of the eye fluid into the subconjunctiva. This operation aims to ensure adequate, long-term, and safe filtration of the aqueous humor from the eye. However, trabeculectomy is associated with a high failure rate in the treatment of NVG, but in combination with anti-antimetabolites such as mitomycin C (MMC) or 5-Fluorouracil (5-FU), the success rate has been improved [60]. MMC is a common antimetabolic agent used after trabeculectomy. It has been shown to inhibit conjunctival/episcleral fibroblast proliferation, thus preventing scarring. Additionally, it produces thinner, avascular, and hypocellular filtration blebs with more atrophic stroma, so better surgical results are obtained [61,62]. Some studies reported that a success rate of trabeculectomy with MMC in the eyes with NVG secondary to PDR was from 62.6% to 81.2% at the end of 1 year; however, after 5 years, it decreased to 51.3% [63,64]. A preoperative intravitreal bevacizumab injection before trabeculectomy has been shown to improve surgical outcomes and reduce the postoperative risk of hyphema [65]. A randomzed, nonblinded, comparative trial by Naveed Nilforushan et al. reported that primary trabeculectomy combined with the intravitreal injection of bevacizumab (2.5 mg) and subconjunctival MMC resulted in decreased IOP compared with subconjunctival MMC alone in patients with diabetes [62].

The previous study Advanced Glaucoma Intervention Study (AGIS) observed that the success rate of trabeculectomy in young patients is poorer than in older individuals with adjunctive MMC [66]. However, another study by Thi Kieu Hau Hoang et al. investigated opposite associations between age and surgical outcomes after trabeculectomy with MMC. A total of 143 eyes were enrolled and divided into two different groups: younger group 1 (below 60 years old) and older group 2 (above 60 years old). The results showed that, compared to the older-age group, younger patients had better long-term surgical outcomes after trabeculectomy with MMC [61].

Glaucoma drainage devices (GDDs). Glaucoma implants are used to increase the IOP-lowering effect by increasing the fluid flow out of the eye. There are two types of GDDs: (1) valved or flow-restrictive implants (Ahmed glaucoma valve (AGV)) and (2) nonvalved implants (Baerveldt, Molteno, and Aurolab aqueous drainage implant (AADI)) [3]. In patients with NVG, valved implants are more commonly used due to their adequate IOP-lowering effect, and the low chances of developing eye hypotony and iris damage after surgery [67]. GDD implantation can be preferred if trabeculectomy fails or there is a high risk of failure because of conjunctival scarring and inflammation [68].

AGV implantation is considered to be effective and safe for the treatment of NVG eyes. The key factors for the success of AVG surgery are age (younger patients considered to be a significant risk factor for surgical failure due to the stronger wound-healing process and fiber-wrap development in the periphery of the drainage disc [69]), standardized retinal photocoagulation history, no postoperative complications, and postoperative anterior chamber hyphema [70,71]. Hyung Bin Hwang et al. observed that AGV with preoperative adjuvant bevacizumab had no significant outcome on decreasing the IOP compared to AGV monotherapy in patients with NVG [72]. On the other hand, preoperative adjuvant bevacizumab is recommended to improve the surgical success rate of AGV implantation [73]. Panos G Christakis et al. compared Ahmed and Baerveldt implants and found that both implants successfully reduced the IOP (in the Ahmed group, the mean IOP was 16.6 ± 5.9 mmHg and 13.6 ± 5.0 mmHg in the Baerveldt group). A lower failure rate was observed in the Baerveldt group [74].

The current scientific evidence comparing surgical methods for the treatment of NVG is limited, and the choice of surgical procedure is primarily based on the patient’s condition and then on the surgeon’s decision. According to the literature, the IOP-lowering efficacy of drainage implants and trabeculectomy is not significantly different, but a trabeculectomy is considered to be a safer procedure [75].

Cyclodestructive procedures are often performed on patients with refractory NVG when other treatments, including GDDs or pharmacological therapy, failed to lower the IOP. The main principle of these procedures is to damage the secretory epithelium of the ciliary epithelium, which leads to a reduction in aqueous humor secretion. The most frequently used procedures are laser cyclophotocoagulation (CPC) and endoscopic cyclophotocoagulation (ECP). Nowadays, a transcleral diode laser (810 nm wavelength) is used to perform CPC [76,77]. Clinical data have shown that both CPC and EPC are safe and effective in controlling the IOP. In Williams et al.‘s study, the success rate of CPC was 75% after 3 months and 66% after 6 months [78]. According to the literature, the IOP-lowering effectiveness of draining implants and CPC does not differ significantly, but draining implants is considered to be safer.

Usually, the treatment options for NVG depend on the stage of glaucomatous damage. In the early glaucoma stage, when the NVI develops, retinal ischemia-reducing treatment (PRP and anti-VEGF) is usually used. Open-angle and closed-angle glaucomas progressing to irreversible damage of the optic nerve are treated similarly, i.e., the treatment of retinal ischemia and correction of high IOP (PRP, anti-VEGF, local and systemic pharmacological treatment, and surgical interventions) [29].

Since NVG is often resistant to any treatment, its progression often leads to blindness and unbearable eye pain. In exceptional cases, when the IOP is already decompensated, the eye is painful and blind, enucleation should be considered [79,80,81].

## 7. Conclusions

NVG is a relatively rare case of secondary glaucomas, but it is aggressively progressing and causing irreversible blindness and ocular pain if left untreated. Ischemic eye diseases (CRVO, PDR, and IOS) are the main contributing factors in NVG pathogenesis which disturb the balance between the angiogenic and antiangiogenic factors, triggering NVI.

NVG is attributed to urgent ophthalmological conditions managed by a team of retinal and glaucoma specialists. The current studies investigating the efficiency of anti-VEGF therapy demonstrate that ranibizumab, bevacizumab, and aflibercept significantly produce the regression of iris neovascularization and result in lower IOP. However, more studies with the latest anti-VEGF inhibitor brolucizumab should be done in the future. Surgical interventions are indicated in advanced stages of NVG when medical treatment does not ensure long-term IOP reduction. The main goal of NVG management is the prevention of angle-closure glaucoma by combining PRP, antiangiogenic therapy, and IOP reduction.

## Data Availability

Not applicable.

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
