# Peer review of "A Review of Neovascular Glaucoma: Etiology, Pathogenesis, Diagnosis, and Treatment"

_medicina, 2022, doi:10.3390/medicina58121870_

Round 1

Reviewer 1 Report

This article discussed the etiology, pathogenesis, diagnosis, and treatment of neovascular glaucoma. There are some issues in this manuscript as follows:

1.    The meaning of the abbreviations should be clearly indicated in the text at their first mention; e.g. VEGF and PRP in the Abstract.

2.    The key words should be simplified as much as possible.

3.     The “Introduction” is too short and doesn’t give the essential background information needed regarding the review subject. Please, revise.

4.    The novel points in this review article should be clarified because there are previous reviews that discussed the same  subject in more details; e.g. https://www.ncbi.nlm.nih.gov/pmc/articles/PMC5116372/, https://www.ncbi.nlm.nih.gov/pmc/articles/PMC6055312/

5.    Page 3: The sentence “NVG is a tremendous ocular condition caused by new blood vessels in the iris and in anterior chamber corner forming in response to retinal ischemia and leading to uncon-trolled IOP rise.” needs a reference. Please, revise

6.    Diagnosis of the “Pre-rubeosis” stage should be mentioned.

7.    Page 7: In VEGF-inhibitors, brolucizumab should be mentioned. It is a humanized monoclonal single-chain variable fragment that binds and inhibits VEGF-A. It was FDA-approved in October 2019 and is considered as the only approved single chain antibody fragment till now.

8.    Page 8: The section of “Glaucoma drainage devices” has only 3 references. Please, enrich this part with more recent references.

9.    In surgical treatment, “minimally invasive glaucoma surgeriesshould be mentioned.

10. In the conclusion, the sentence “A key role in the pathogenesis of NVG plays ischemic eye diseases” should be revised and rephrased.

11. I think that the conclusion was insufficient. It should identify the possible clinical implications of the data obtained from the present review.

12. The “references” should be written in the same format of the journal “Medicina”.

13. The manuscript should be thoroughly checked regarding the grammatical and typing errors.

Reviewer 2 Report

1.    No doubt it is a good literature review regarding the Neovascular glaucoma (NVG), its prevalence and etiology. I recommend the authors must include their own opinion in each section of this manuscript.

2.    The introduction part is very short; it can be elaborated further with new citations of recently published articles in this line of research works. 

3.    In Table 2. Clinical stages of NVG, insert a column at right most side for references and for each stage mention one or more reference(s). Similarly, for Table 1 also.

4.    Some abbreviations like VEGF and PRP appeared first time in this manuscript in the abstract section should be elaborated before and put the abbreviations in the parentheses.

5.    In section 5. Opthalmic diagnostics, correct the spelling of “Ophthalmics”. Also in the same section where the “NVG patients should be carefully investigated because of the systemic associations:” described, for each description puta separate reference just after the end of each statement of diagnosis.

6.     Double check the spelling mistakes throughout the manuscript for example, prowen should be “proven” opthalmic should be ophthalmic and so on.

7.    Treating the underlying pathology, the authors should go for further available literatures regarding the use of drugs other than only mentioned in this section (Anti-VEGF inhibitors like bevacizumab and ranibizumab intravitreal injections).

8.    What was the rout or dosage form of aflibercept monotherapy used in Japan? Provide information about the available dosage forms with the manufacturers details at this point. Similarly, details for the bevacizumab and ranibizumab intravitreal injections availability in the market.

9.    Double check the statement “Key factors for the success of AVG surgery are age (younger than ≤55 years considered to be significant risk factor of surgical failure)”.

10. “Current scientific evidence comparing surgical methods for the treatment of NVG is limited, and the choice of surgical procedure is primarily based on surgeon’s decision and assessment of the patient's condition” rather its primarily based on the patient's condition and then surgeon’s decision.

11. The authors should discuss little about the adjunctive therapy (by mitomycin C) after trabeculectomy considering the relationship between age and surgical success after trabeculectomy.

Round 2

Reviewer 1 Report

The authors had appropriately addressed my comments

Reviewer 2 Report

Authors have revised the manuscript as per the suggestion.